# Mono-CYP CHO Model: A Recombinant Chinese Hamster Ovary Cell Platform for Investigating CYP-Specific Tamoxifen Metabolism

**DOI:** 10.3390/ijms26093992

**Published:** 2025-04-23

**Authors:** Christian Schulz, Sarah Stegen, Friedrich Jung, Jan-Heiner Küpper

**Affiliations:** 1Fraunhofer Institute for Cell Therapy and Immunology, Branch Bioanalytics and Bioprocesses (IZI-BB), Site Lausitz (IZI-BB-L), 01968 Senftenberg, Germany; sarah.stegen@izi-bb.fraunhofer.de; 2Institute of Biotechnology, Brandenburg University of Technology Cottbus-Senftenberg, 01968 Senftenberg, Germany; friedrich.jung@b-tu.de (F.J.); institut-biotechnologie@b-tu.de (J.-H.K.)

**Keywords:** Chinese hamster ovary cells, mono-CYP CHO, liver, phase-1 biotransformation, cytochrome P450 monooxygenase, CYP2D6, tamoxifen, endoxifen, *N*,*N*-didesmethyltamoxifen, DDM-TAM

## Abstract

The metabolism of drugs and foreign substances in humans typically involves multiple enzymatic steps, particularly in phase-1 biotransformation in the liver, where various cytochrome P450 monooxygenases (CYPs) play crucial roles. This complexity can lead to a wide range of metabolites. Understanding the contributions of individual CYPs and their interactions within these intricate enzyme cascades can be challenging. We recently developed an in vitro biotransformation platform employing various Chinese Hamster Ovarian (CHO) cell clones. These clones express human cytochrome P450 oxidoreductase (CPR), and each is defined by a specific human CYP enzyme expression, thus exhibiting no detectable endogenous CYP enzyme activity (mono-CYP CHO platform). In this study, we investigated whether the mono-CYP CHO platform is a suitable tool for modeling complex drug metabolization reactions in vitro. Tamoxifen (TAM) was selected as a model substance due to its role as a prodrug widely used in breast cancer therapy, where its main active metabolite, endoxifen, arises from a two-step metabolism primarily involving the CYP system. Specifically, the combined activity of CYP3A4 and CYP2D6 is believed to be essential for efficient endoxifen production. However, the physiological metabolization pathway of TAM is more complex and interconnected, and the reasons for TAM’s therapeutic success and variability among patients are not yet fully understood. Analogous to our recently introduced mono-CYP3A4 CHO cells, we generated a CHO cell line expressing human CPR and CYP2D6, including analysis of CYP2D6 expression and specific activity. Comparative studies on the metabolization of TAM were performed with both mono-CYP CHO models individually and in co-culture with intact cells as well as with isolated microsomes. Supernatants were analyzed by HPLC to calculate individual CYP activity for each metabolite. All the picked mono-CYP2D6 clones expressed similar CYP2D6 protein amounts but showed different enzyme activities. Mono-CYP2D6 clone 18 was selected as the most suitable for TAM metabolization based on microsomal activity assays. TAM conversion with mono-CYP2D6 and -3A4 clones, as well as the combination of both, resulted in the formation of the expected main metabolites. Mono-CYP2D6 cells and microsomes produced the highest detected amounts of 4-hydroxytamoxifen and endoxifen, along with *N*-desmethyltamoxifen and small amounts of *N*,*N*-didesmethyltamoxifen. *N*-desmethyltamoxifen was the only TAM metabolite detected in notable quantities in mono-CYP3A4, while 4-hydroxytamoxifen and endoxifen were present only in trace amounts. In CYP2D6/3A4 co-culture and equal mixtures of both CYP microsomes, all metabolites were detected at concentrations around 50% of those in individual clones, indicating no significant synergistic effects. In conclusion, our mono-CYP CHO model confirmed the essential role of CYP2D6 in synthesizing the active TAM metabolite endoxifen and indicated that CYP2D6 is also involved in producing the by-metabolite *N*,*N*-didesmethyltamoxifen. The differences in metabolite spectra between the two mono-CYP models highlight the CYP specificity and sensitivity of our in vitro system.

## 1. Introduction

Drugs and foreign substances are metabolized in humans by a large number of enzymatic reactions. These reactions can influence each other and are partly interdependent. This is particularly evident in phase-1 biotransformation in the human liver, where multiple cytochrome-P450 monooxygenases (CYPs) and other enzymes, such as carboxylesterases, are involved. To a lesser extent, flavin-containing monooxygenases and monoamine oxidases also play a role in drug metabolization. This can lead to a broad spectrum of metabolites [1,2]. Depending on the drug, these may have further desirable or undesirable physiological activities. The complexity of phase-1 biotransformation sometimes makes it difficult to elucidate the influence of individual CYPs on drug conversion. In addition, it is difficult to understand the interactions between them in the context of complex enzyme cascades in physiological in vitro models such as human liver cells or microsomes.

In the past, a variety of in vitro systems have been developed to simulate liver metabolism and to understand the metabolization of drugs and xenobiotics as well as interactions between them in the human organism. These systems include pre-cut liver slices, cellular 2D/3D and organ-on-a-chip models from primary, primary-like (e.g., Upcyte^®^ hepatocytes) and, increasingly used in recent years, iPSC-derived hepatocytes as well as cell lines of carcinogenic origin such as HepG2 and HepaRG. Subcellular fractions derived from hepatocytes or directly from liver tissue such as cytosol S9 or microsomes, respectively, are also well-established test models in drug research and toxicology [3,4,5]. Between all these systems, there are often significant variations found in the expression of phase-1 and phase-2 enzymes and in the enzymes’ lifespans [3]. Nevertheless, all these models have one key thing in common: due to their origin, they all have a spectrum of biotransformation activities to a greater or lesser extent [6,7,8,9]. This allows the physiological situation to be simulated in vitro. However, it is more difficult to gain specific insights into which phase-1 enzymes, particularly CYPs, are involved in drug metabolization and which metabolites they generate. Likewise, drug effects on specific CYPs and CYP–CYP interactions are difficult to clarify in systems with more than one CYP activity. However, it is clear that the use of recombinant humanized mono- and multi-CYP models, including the expression of corresponding redox partners such as cytochrome P450 oxidoreductase (CPR) and cytochrome b5 (Cyt-B5), helps to answer such questions. These models are frequently used to provide supplementary evaluations of results from more complex liver models [5,10]. They did provide some insights into human metabolic clearance, including CYP profiling, CYP-specific metabolite spectra, and CYP–CYP and CYP–drug interactions. In vitro systems for studying liver-specific phase-1 biotransformation with heterologous expression of recombinant human CYPs are diverse and based on bacterial, yeast, or mammalian cells as well as baculovirus-driven expression in insect cells (i.e., *Sf*21) [10,11,12,13,14]. The latter is primarily used for the supply of microsomes with specific CYP activity, which are available as so called “Supersomes^®”^ or “Baculosomes^®”^ [9,10,15]. However, it is important to note that due to the induction of recombinant protein expression by viral infection, long-period cell-based in vitro studies, i.e., on the cell mobility of a substance, are more difficult to implement [16,17,18]. Furthermore, complex post-translational modifications can differ from those in mammals, potentially limiting protein function. Additionally, endogenous CYP enzyme activities in these systems are not well characterized.

Our in vitro model, which we presented recently, successfully addresses the above challenges. It is based on Chinese Hamster Ovary cells (CHO-K1, hereafter referred to as CHO), that have been recombinantly and stably equipped with human CPR in the first step. In a second step, human cDNAs for various CYP enzymes were transduced by lentiviral gene transfer [19]. CHO cell models have been established for studies on human liver metabolic disorders since the 1990s [20,21,22]. Recombinant CYP-expressing CHO cells are perfectly suited for phase-1 biotransformation studies as no endogenous CYP activities have been detected to date [20,21,23]. According to our own findings, carboxylesterase activity also appears to be very low compared with liver cells and microsomes (unpublished data). The serum-free cultivation of CPR/CYP-modified CHO cells (hereinafter referred to as mono-CYP CHO) in suspension opens up the possibility of scaling and facilitates their potential to produce metabolites for various applications (e.g., preclinical drug development). However, the current focus is more on the validation and further exploration of potential applications of the mono-CYP CHO platform. Generating high biomass compared with conventional 2D cultures enables the realization of sensitive cell-based metabolization studies for a longer period. Additionally, it allows for the preparation of microsomal fractions (MFs) with specific CYP activity from the same cell clone, facilitating comparability and reproducibility. In particular, microsomes are a widely used high-throughput in vitro screening model in pharmacology for phase-1 liver metabolism studies [4]. Obtained by subcellular fractionation, they represent membrane-containing enrichments of the endoplasmic reticulum and thus of CYP enzymes (including co-enzymes) from cells or tissues [24]. Unlike the cytosolic and S9 fractions, human liver microsomes (HLMs) isolated from liver cells contain only a few phase-2 (predominantly uridine 5′-diphospho-glucuronosyltransferases, UGTs) and other metabolically active cytosolic enzymes such as esterases, amidases, or epoxide hydrolases [25]. This makes it impossible to fully reproduce the liver metabolism, but it also prevents the further metabolization of CYP metabolites to a significant extent.

The selection of a suitable model substance was based on the questions to be addressed with the mono-CYP CHO platform. Our test models primarily aim to elucidate drug metabolization in the context of phase-1 biotransformation studies on liver metabolism. This includes CYP profiling, CYP-specific metabolite spectra, as well as CYP–CYP and CYP–drug interactions. Accordingly, we selected tamoxifen (TAM) as a model substance because it is of high clinical relevance. It is a key advantage that TAM’s physiological biotransformation profile is predominantly mediated by the CYP system.

TAM is an anti-cancer drug that has been used since the 1970s to treat early and metastatic breast cancer. It belongs to the class of selective estrogen receptor modulators (SERMs) [26]. Its effectiveness is proven with a reduction in recurrence and mortality rates of up to 50% and 30%, respectively. TAM is considered an essential component of endocrine therapy and the prevention of pre- and postmenopausal and male estrogen receptor (ER)-positive breast cancer [27,28]. Phase-1 bioactivation in the liver by the CYP system is required to develop its full anti-carcinogenic effect [29,30]. TAM is used clinically as an estrogen receptor (ER) antagonist. It competes at the cellular level with estrogen for ER-positive cancer cells, and estrogen is necessary for these cells to proliferate [31,32,33]. Furthermore, TAM also has pro-apoptotic effects on cells via ER-independent signaling pathways [34].

TAM is undoubtedly a prodrug, and its liver-formed metabolites are the main drivers of its therapeutic success due to their significantly stronger anti-estrogenic effects on breast cancer cells [10,29,35,36]. Originally, (Z)-4-hydroxytamoxifen (4-OH-TAM), a main metabolite formed by CYPs, was assumed to be the active metabolite, as it has a ~100-fold higher affinity to the ER compared with TAM [10,35]. However, (Z)-4-hydroxy-*N*-desmethyltamoxifen (endoxifen), which is also formed by CYPs (especially by CYP2D6), has a similarly high ER affinity [37,38,39]. However, there are large differences in plasma concentrations between the two metabolites in patients with normal pharmacological distribution and elimination. Clinical studies have shown concentrations of endoxifen up to 10-fold higher, with large variations between patients [29,39,40,41,42]. Endoxifen is the only known TAM metabolite that not only inhibits the ER but also accelerates the proteasomal degradation of ERα [43]. Therefore, according to current knowledge, it is assumed that endoxifen is the main active metabolite.

The biotransformation of TAM in the human liver is a highly complex network of metabolizations. A large number of different enzymes catalyze this process (Figure 1). This is largely dominated by multiple CYP enzymes but also UGTs, sulfotransferases (SULTs), and possibly other biotransformation enzymes. Various metabolites are formed, some of which have further pharmacological effects [29]. In vitro studies have shown that some TAM metabolites, especially norendoxifene, act as strong inhibitors of aromatase [44]. Physiologically, for example, this would be accompanied by a reduction in estrogen levels, which would increase the anti-proliferative effect on ER-positive breast cancer cells. The observed therapeutic success is undoubtedly the result of cumulative effects of different TAM metabolites.

CYPs are the determining factor in the pharmacokinetics of TAM, with 4-hydroxylation and *N*-demethylation being the major metabolic pathways to the most important main metabolites. The most effective metabolite, endoxifen, is formed from both pathways in a second metabolic reaction. The 4-hydroxylation pathway is catalyzed by several CYPs, with 4-OH-TAM being formed as a result. CYP2D6 is the main contributor here, but other CYPs such as CYP2B6, -2C9, -2C19, and -3A4/5 are also involved [29,30]. From a physiological perspective, however, this reaction pathway accounts for only approximately 7% of TAM metabolism. In contrast, the demethylation of TAM to *N*-desmethyltamoxifen (DM-TAM) constitutes ~92% of its physiological fate and is catalyzed primarily by CYP3A4/5 [10,36,42,46]. CYP1A1, -1A2, -2C9, -2C19, and -2D6 are also involved [29,30,47], but the role of CYP2D6 is less certain [26]. In subsequent metabolizations, DM-TAM is oxidized to several minor metabolites. These include *N*,*N*-didesmethyltamoxifen (DDM-TAM) and 4′-hydroxy-N-desmethyltamoxifen (4′-OH-DM-TAM). According to current knowledge, the formation of DDM-TAM seems to be exclusively attributed to CYP3A4/5, and the formation mechanism of 4′-OH-DM-TAM still is unknown [29,30,45]. Endoxifen emerges from both pathways. It is produced by further hydroxylation of DM-TAM, exclusively by CYP2D6, and by demethylation of 4-OH-TAM, primarily by CYP3A4 but also to a lesser extent by CYP2C19, -2D6, and -3A5 [29,36,42]. In the course of further deactivation and preparation for excretion, various UGTs or SULTs catalyze the glucuronidation and sulfation of TAM and its metabolites [48,49]. Physiologically, excretion takes place primarily in the form of glucuronides via the bile [36].

In addition to liver-based models such as human hepatocytes and liver microsomes, recombinant CYP systems, mostly in the form of microsomes, have also been used for the in vitro elucidation of the TAM phase-1 biotransformation. In most cases, recombinant human CYPs expressed in baculovirus-infected insect cells in the form of microsomes were used in those studies. However, additional research efforts are needed to fully understand the pharmacokinetics of TAM. Some TAM metabolites, such as 3-hydroxytamoxifen and 4′-hydroxy-*N*-desmethyltamoxifen, have so far been detected only in vitro or in animal models, but not in patients [50,51].

The objective is to determine whether they induce physiological effects and, if so, to understand the mechanisms involved. Not all metabolites have been fully characterized, and some of the enzymes responsible for their synthesis are still unknown. Recombinant expression systems such as mono-CYP CHO, with high and specific CYP enzyme activity, offer a valuable approach to gaining insights into this field. These systems make it possible to simulate complex metabolization pathways with single or multiple CYP involvement in vitro and come a step closer to answering unanswered questions. With regard to TAM, for example, it is still not clear whether CYP2D6 or possibly other CYP enzymes contribute to the formation of the side metabolite DDM-TAM or whether this is realized solely by CYP3A4/A5 [26,29,30]. The same applies to 3-hydroxytamoxifen (Droloxifene^®^), another potent SERM with pharmacokinetics different from TAM, but which has not yet been pursued after phase-II and -III clinical trials in 2000 [52,53].

We have intentionally excluded further description and classification of the pharmacogenomic influences on the biotransformation of TAM. This decision was made to avoid adding an unacceptable level of complexity to the TAM model. For more comprehensive insights on this topic, we refer to some reviews [29,54,55].

Based on our classifications, it becomes clear that TAM emerges as a well-suited model substance for further validation of the mono-CYP CHO platform. Its high clinical relevance and the fact that its complex physiological metabolization is significantly influenced by the CYP system predestines this prodrug for studies with our cellular and microsomal models. Despite the intensive cumulative research efforts of the past decades, the complex network of diverse biotransformation enzymes in TAM metabolization is a chapter that is still not closed.

## 2. Results

### 2.1. Recombinant CPR/CYP2D6 Expression and Activity in Selected Mono-CYP2D6 CHO Clones

After having transduced lentiviral CYP2D6 cDNA into the previously generated CHO-CPR C12 clone, we were able to isolate 19 individual clones [19]. Recombinant CPR/CYP2D6 expression was initially characterized at the protein level by immune detection and cell-based GloCYP2D6 activity assay (Figure 2A,B). For CHO-CPR C12 and all mono-CYP2D6 CHO clones, a distinct band for recombinant human CPR around 80 kDa was detected, which was much weaker for parental CHO cells and HepG2-CYP2D6. CYP2D6 expression was detectable as a specific protein band at 55 kDa for all cell lines. Whereas no significant difference in the CYP2D6 band intensity could be detected between the mono-CYP2D6 CHO clones, a band of the expected size was weaker but clearly detectable in CHO-CPR C12 and the parental CHO cells. HepG2-CYP2D6 was comparable to the mono-CYP2D6 CHO clones with regard to CYP2D6. All cell lines showed a similar loading control GAPDH intensity, with the exception of HepG2-CYP2D6.

Since the sensitivity of immunodetection was not sufficient to detect possible differences in CYP2D6 protein expression, and no conclusions could be drawn about the presence of functional CYP2D6 enzyme, further characterization of the mono-CYP2D6-CHO clones was performed by determining CYP2D6 activity at the level of intact cells. This analysis revealed a heterogeneous CYP2D6 activity in the clone population analyzed (Figure 2B). In comparison, some clones showed higher CYP2D6 activity than others, while no CYP2D6 activity could be detected in CHO-CPR C12 and the parental CHO cells. Surprisingly, HepG2-CYP2D6 exhibited much lower CYP2D6 activity compared with the mono-CYP2D6 clones. Based on these results, the three mono-CYP2D6 CHO clones C1, C9, and C18 with the highest CYP2D6 activity were selected for further characterization at the microsomal level. Here, differences in CYP2D6 activity were also detectable (Figure 2C). With an enzyme activity of 3.17 ± 0.31 pmol/mg_protein_×min, C18 showed the highest activity, followed by C9, with 2.61 ± 0.35 pmol/mg_protein_×min, and C1, with 2.14 ± 0.15 pmol/mg_protein_×min. Compared with mono-CYP2D6 CHO C18, the metabolization of the luminogenic CYP2D6 substrate luciferin-ME EGE into luciferin was higher in commercially available human liver microsomes (HLMs) as “physiological reference” with 4.75 ± 0.33 pmol/mg_protein_×min. In contrast, no formation of luciferin could be detected with CHO-CPR C12, which served as the parental control.

### 2.2. Characterization of Isolated Mono-CYP CHO Microsomes (CYP2D6 and CYP3A4)

Prior to the metabolization studies with TAM, microsomal fractions (MFs) of mono-CYP2D6 CHO C18 and mono-CYP3A4 CHO C1 were characterized by Western Blotting to validate the isolation protocol in comparison with the parental CHO cell line and HLM as the physiological reference at the protein level (Figure 3). Isolated MFs of the mono-CYP CHO models for CYP2D6 and CYP3A4, which are both derived from the CHO-CPR 12 clone, showed a higher amount of CPR compared with CHO MFs and a higher amount compared with HLMs. Regarding CYP3A4, HLMs showed higher protein levels than MFs from mono-CYP3A4 CHO C1. No CYP3A4 was detected in MFs from parental CHO cells. The situation was different for CYP2D6, where low protein levels of basally expressed CYP2D6 were detected in CHO MFs (right blot in Figure 3). A clear CYP2D6 protein band was also found in HLMs. However, mono-CYP2D6 CHO MFs showed a significantly stronger CYP2D6 protein band compared with both references. Protein detection of VDAC-1, a mitochondrial marker, revealed higher amounts in HLMs compared with MFs from CHO cell lines. In all MFs from CHO cells, a distinct protein band for VDAC-1 could be detected, and no notable difference between parental CHO and mono-CYP CHO was visible.

### 2.3. Metabolization of Tamoxifen with Mono-CYP CHO Cells

To investigate the CYP-specific metabolization of TAM by mono-CYP CHO models, metabolization studies were performed with intact suspension cells and isolated MFs. In the course of the metabolization studies with the cells, a slight pre-existing contamination of the substrate with about 0.3% DM-TAM was detected. This remained unchanged during the incubation of TAM-blanks (BG-TAM), which contained only TAM, but no cells, in the buffer system used. In addition, no further TAM metabolites were detected in the TAM-blanks, neither at the beginning nor after 4 h of incubation. Nevertheless, it is striking that significantly less TAM than the originally added 100 µM was detectable in the supernatant of TAM-blanks after 4 h than at the beginning (Figure 4A). After metabolization, minimal amounts of DM-TAM and 4-OH-TAM were detected in the supernatants of parental CHO cells above the background of TAM-blanks. Surprisingly, the original concentration of TAM did not decrease significantly over time, unlike in the TAM-blanks. Endoxifen could not be detected in the supernatants from parental CHO cells. After metabolization of TAM with the mono-CYP CHO models, a significant reduction in detectable TAM was observed in both the mono- and the co-culture compared with CHO parental cells. Mono-CYP2D6 CHO showed the highest reduction and mono-CYP3A4 CHO the lowest (Figure 4A). The co-culture was close to the level of mono-CYP2D6 CHO. DM-TAM was detected in the supernatants of both mono-CYP models individually and in co-culture. Mono-CYP3A4 CHO showed the highest concentration, followed by the co-culture and mono-CYP2D6 CHO. The detected concentrations were several times higher with all mono-CYP CHO samples than in parental CHO cells. Notable concentrations of 4-OH-TAM were detected solely in supernatants of mono-CYP2D6 CHO and in the co-culture. Mono-CYP3A4 CHO showed very low amounts at the level of the parental cells, close to the detection limit. A similar picture emerged in the detection of the TAM active metabolite endoxifen from culture supernatants. Solely in mono-CYP2D6 CHO and the co-culture, significant amounts of endoxifen were detected. The endoxifen concentration in mono-CYP2D6 CHO was almost twice as high as in the co-culture. This shows that CYP2D6 as a single enzyme, in contrast to CYP3A4, is capable of generating endoxifen from TAM and that in our cell-based in vitro model, there were no recognizable synergistic effects between the two CYP enzymes.

The data collected were used to calculate the CYP enzyme activities for the formation of the TAM metabolites (Figure 4B). For the calculation, the data were normalized to the protein level and, in the case of DM-TAM, corrected for the detected background in TAM-blanks. The formation rates of both main metabolites 4-OH-TAM and DM-TAM in the cell suspensions differed significantly between the CHO models used. The mono-CYP2D6 CHO showed the highest formation rate for 4-OH-TAM followed by the co-culture, whereas the formation rate in mono-CYP2D6 CHO was about 1.4-fold higher than in the co-culture. In contrast, the rate of 4-OH-TAM formation in mono-CYP3A4 CHO and the parental cells was close to zero. For DM-TAM, the picture was different. Corrected from the background of the TAM-blanks, this main metabolite was formed in mono-CYP3A4 CHO alone about 1.5-fold faster than mono-CYP2D6 CHO and about 1.3 times faster than in co-culture. Compared with the mono-CYP CHO models, a low DM-TAM synthesis rate was also calculated for parental CHO cells. Significant enzyme activity in the formation of the main active metabolite endoxifen was detected only in mono-CYP2D6 CHO and the co-culture, with mono-CYP2D6 CHO forming endoxifen approximately 1.5 times faster. Endoxifen formation was almost zero in mono-CYP3A4 and non-existent in the parental CHO cells.

In addition to the main metabolites mentioned, weaker signals indicating the formation of side metabolites were also detected in the spectra. These were detectable only in the mono-CYP CHO cells and often showed only slightly altered retention times in relation to the main metabolites. In addition to several signals that were too close to the detection limit for adequate quantification, one metabolite signal stood out particularly clearly in the mono-CYP2D6 CHO and the co-culture (Figure 5A). This metabolite could be identified as *N*,*N*-didesmethyltamoxifen (DDM-TAM) by additional mass spectrometric characterization using LC-MS. Due to the lack of a corresponding standard substance, only the signal intensity between the different cell models was compared in this study (Figure 5B). Mono-CYP2D6 CHO showed the strongest signal, with a peak area of A_mono-CYP2D6 CHO_ = 7196 ± 336, followed by the co-culture, with A_co-culture CYP2D6/3A4_ = 4405 ± 695. In mono-CYP3A4 CHO, the signal was significantly weaker, with A_mono-CYP3A4 CHO_ = 603 ± 177.

### 2.4. Metabolization of Tamoxifen with Mono-CYP CHO Microsomal Fractions

In contrast to intact cells, a period of TAM metabolization between 2 and 6 h was implemented in the microsomal studies in order to include kinetic aspects of the mono-CYP CHO models. Since the focus was on the detection of the active metabolite endoxifen, shorter conversion times were not useful, as endoxifen could be clearly detected in HLMs for the first time after more than 2 h (Figure 6B). The slight pre-existing contamination of the commercially obtained substrate TAM with DM-TAM previously observed in the cell studies amounted to an average of 1% in the experiments with MF and did not change over the test period.

When monitoring the substrate TAM over the entire metabolization period, an interesting observation was made. In the HLM samples and the TAM-blanks, significantly lower TAM concentrations were always detectable compared with samples with CHO MFs of any kind (Figure 6A). This was true for both samples with and without NADPH supplementation. In the TAM-blanks and HLMs without NADPH addition, only about 40–60% of the originally added 100 µM TAM was detectable after 2 h. Regardless of NADPH, in parental CHO MFs, it was on average around 80–90%. Additionally, only TAM-blanks without the addition of microsomes showed a tendency for further reduction in TAM over time. For all other samples, the fluctuations in the TAM quantification were too large to draw any conclusions.

Likewise, similar to the cell-based experiments, clear differences in the type and amount of TAM metabolites formed between the different MFs were recognizable. An interesting observation was that after 2 h of metabolization, none of the mono-CYP CHO models used resulted in further formation of metabolites over time, and the detected metabolite concentrations remained almost constant. Accordingly, the formation rates for the investigated TAM main metabolites were very low or close to zero in the period after 2 h (Figure 6C). The situation was different for the HLMs. For the main metabolites, an increase in the amounts, but not in the formation rate, could be detected in the course of metabolization up to 6 h. An exception was the formation rate of endoxifen, which reached its maximum at 4 h. In the controls of HLMs and the MFs of CHO parental cells without addition of the NADPH regeneration system, no metabolite formation could be detected at any time (Figure 6B). In parental CHO MFs with NADPH, the formation of a small amount of DM-TAM was detected, which was close to that also found in the TAM-blanks. 4-OH-TAM and endoxifen were not detected.

A detailed comparison of the TAM metabolites formed by different MFs shows that only DM-TAM was detectable in all models examined and that its concentrations were of approximately the same order of magnitude (Figure 6B). Both mono-CYP CHO MFs and co-CYP2D6/-3A4 MFs were constant during the metabolization period. HLMs, on the other hand, showed an accumulation of DM-TAM over time despite a steady reduction in the formation rate, and they reached approximately the level of mono-CYP3A4 MFs after 6 h. Looking at the initial sampling time, approximately twice as much DM-TAM was generated within the first 2 h in mono-CYP2D6 MFs than in HLMs. 4-OH-TAM, as the other main TAM metabolite that can lead to endoxifen, was formed exclusively by those mono-CYP CHO MF samples containing CYP2D6. Mono-CYP2D6 MFs had a two-fold higher formation rate than co-CYP2D6/-3A4 MFs. Only traces of 4-OH-TAM could be detected in mono-CYP3A4. In direct comparison with HLMs, a ~14-fold higher concentration of 4-OH-TAM was initially (after 2 h) detectable in mono-CYP2D6 MFs and ~7-fold higher in co-CYP2D6/-3A4 MFs. Similar to DM-TAM, an increase in the concentration of 4-OH-TAM over time was recognizable in HLMs, whereby the formation rate steadily decreased. When detecting the formation of the TAM active metabolite endoxifen, the picture was basically the same as for 4-OH-TAM. Only in MFs with mono-CYP2D6 was endoxifen reliably detectable, whereas again in mono-CYP2D6 MFs, an endoxifen concentration about twice as high as in co-CYP2D6/-3A4 MFs was detected. In direct comparison with HLMs, a ~40-fold higher concentration of endoxifen was initially (after 2 h) detectable in mono-CYP2D6 MFs and ~16-fold higher in co-CYP2D6/-3A4 MFs. As with the previously considered TAM metabolites, an increase in the concentration of endoxifen over time was recognizable in HLMs during formation, whereby, unlike before, the formation rate was at maximum at 4 h (Figure 6C).

Comparable to the cell-based experiments, other low-intensity metabolite peaks were also detected after TAM metabolization in mono-CYP CHO MF models in addition to the main metabolites mentioned. Close to the retention time of 4-OH-TAM, two further metabolite peaks with lower intensities were always detected in mono-CYP2D6 and co-CYP2D6/-3A4. The intensities for mono-CYP2D6 MFs were about twice as high as for co-CYP2D6/-3A4 MFs. With HLMs, these signals were only sporadically detectable, with intensities close to the detection limit. DDM-TAM, as another side metabolite with slightly shorter retention time than DM-TAM also observed during cellular experiments, was more clearly detectable and evaluable (Figure 6B). It could be detected in mono-CYP2D6, co-CYP2D6/-3A4, and HLMs. The peak intensities were almost constant over the entire period. This metabolite was not detected in MFs of parental CHO and, interestingly, also not in mono-CYP3A4 CHO.

## 3. Discussion

In this study, we investigated the reliability of our novel mono-CYP CHO platform, which features recombinant expression of human CPR and specific CYP enzymes, for modeling complex drug metabolization in vitro. The prodrug tamoxifen (TAM) was chosen as a model substance due to its clinical significance in breast cancer therapy. Despite decades of research, the CYP-dominated metabolization pathway in the human liver remains not fully understood, making TAM an ideal candidate for our studies. Our work aims to enhance the understanding of TAM’s therapeutic success and its variability among patients. We focused on synthesizing TAM’s major phase-1 metabolites and the formation of the main therapeutically active metabolite endoxifen. For this purpose, we utilized two mono-CYP CHO models with specific CYP activity of CYP2D6 or CYP3A4. Metabolization studies were conducted with intact cells and isolated microsomes from both systems as well as in equivalent mixtures. We monitored the metabolites formed and the concentration of the substrate TAM in the in vitro models using HPLC analytics.

The first step in establishing our test model to investigate the CYP-dependent metabolization of TAM and validate the mono-CYP CHO in vitro platform was generating the necessary missing mono-CYP CHO cell models. We focused on two key CYP enzymes, CYP2D6 and CYP3A4, involved in forming the active metabolite endoxifen [26,29,30]. Our mono-CYP3A4 CHO cell model has been published [19], but mono-CYP2D6 cell clones still needed to be generated and characterized. After lentiviral gene transfer of human CYP2D6 cDNA into the CHO-CPR C12 cell line and antibiotic selection, we compared the recombinant expression of CYP2D6 in 19 isolated clones with that of the parental CHO and CHO-CPR C12 line. All clones showed strong expressions of human CPR and CYP2D6, making it difficult to identify the best clone (Figure 2A). Weak basal expression of CPR and CYP2D6 was detected in the parental lines, but subsequent studies did not reveal basal CYP2D6 enzyme activity in either CHO or CHO-CPR C12 cells at the cellular or microsomal level (Figure 2B,C). To ensure functional CYP2D6 holoenzyme was present in the mono-CYP2D6 clones, we performed a cell-based CYP2D6 activity study using the GloCYP2D6 assay, as we could not isolate microsomal fractions (MFs) from all clones (Figure 2B). Although the manufacturer’s instructions indicate that this test is not suitable for intact cells due to the low membrane permeability of the substrate luciferin-ME EGE, we were able to detect the CYP2D6 activities in the individual clones with sufficient precision. This allowed us to select the three most active clones, C1, C9, and C18, for MF isolation. In subsequent activity determination in MFs using the same assay, parental CHO-CPR C12 and human liver microsomes (HLMs) served as the appropriate physiological reference. C18 was identified as the most suitable mono-CYP2D6 clone, with 3.17 ± 0.31 pmol/mg_protein_×min, and it was selected for the metabolization studies with TAM (Figure 2C). In comparison, HLMs showed about 50% higher CYP enzyme activity. According to the manufacturer, the substrate luciferin-ME EGE is not CYP2D6-specific and is also metabolized into luciferin by CYP1A1, -1A2, and -2B6. Unlike the mono-CYP CHO, HLMs possess a broad spectrum of CYP enzymes [4,10,24]. Thus, these additional enzymes likely contribute to the detected activity. In contrast, the activity observed in mono-CYP2D6 clones is likely solely due to recombinant CYP2D6, as CHO-CPR C12 showed no activity.

Further characterization before the metabolization studies with TAM focused on detecting recombinant CPRs and CYPs in cell-derived MFs. This validated the microsomal isolation protocol, allowing us to compare the performance of mono-CYP CHO MFs with HLMs and the corresponding cell models. Successful isolation of endoplasmic reticulum microsomes occurred in both mono-CYP CHO models, revealing larger amounts of CPR and the respective CYP enzymes. Notably, CPR levels were higher than those in HLMs, and CYP2D6 levels in mono-CYP2D6 MFs also exceeded those in HLMs. Conversely, CYP3A4 was found in higher amounts in HLMs than in mono-CYP3A4 MFs. Assuming a correlation between protein quantity and activity, this aligns with supplier data of the HLMs, indicating higher CYP3A4 activity compared with CYP2D6 (V_max-CYP2D6_ = 0.03 pmol/mg_protein_×min; V_max-CYP3A4-6βOH-T_ = 3.74 pmol/mg_protein_×min; HMMCPL-Lot: PL050J). Similar to previous characterizations, parental CHO MFs exhibited no CYP3A4 and low CYP2D6 protein levels. VDAC-1, detected as a mitochondrial marker protein and thus quality control for the successful isolation of microsomal membrane fragments of the endoplasmic reticulum, was significantly lower in CHO-based MFs than in HLMs, likely due to differences in isolation protocols or expression levels between human and hamster. Importantly, the VDAC-1 levels in mono-CYP CHO MFs were significantly lower than those of the recombinantly expressed proteins.

To adequately discuss the results of the TAM metabolization studies, the mono-CYP CHO platform must be contextualized within the previous research on TAM metabolism. Studies often use patient material, such as plasma and tissue samples from breast cancer therapy, to investigate TAM metabolism [28,29,37,42]. Such studies have the advantage of reflecting the full spectrum of physiological processes and the clinical situation. Previous research has provided insights into how treatment success depends on CYP2D6 gene polymorphisms [29]. However, it is difficult to isolate individual reaction steps and dependencies in detail due to the high complexity and diversity of processes in the human body, some of which take place in parallel and influence each other. Additionally, detecting very small quantities of physiologically formed metabolites ex vivo can be difficult [50,51]. Recombinant in vitro models, such as microsomes (i.e., Baculosomes^®^, Supersomes^®^), including the mono-CYP CHO platform, provide options to address unanswered questions not feasible in clinical trials [9,10,15,51]. Increased activity of specific CYP enzymes from overexpression, combined with low basal CYP activity and restricted metabolic capabilities, facilitates more sensitive detection of metabolites and the controlled investigation of metabolic dependencies. These in vitro systems allow for targeted studies of CYP-dependent reactions (CYP profiling), CYP-specific metabolite elucidation (Met-ID), CYP–CYP interactions, and CYP inhibition, which are crucial for drug validation and safety assessment. The mono-CYP CHO platform also enables the integration of cellular and microsomal studies, as they originate from the same cell clone.

The initially detected contamination in TAM, with ~0.3% DM-TAM in the cellular and ~1% in the microsomal system, was unexpected but within the manufacturer’s quality criteria, guaranteeing ≥ 98% purity (HPLC-grade). The detection difference between the two systems can be attributed to the higher TAM concentration of 100 µM used in microsomal studies, while TAM was limited to 50 µM in the cell system due to concentration-dependent toxic effects. The higher TAM concentration, and thus DM-TAM content, led to more accurate quantification, helping to avoid misleading conclusions about substrate instability or false positive metabolization in sample preparations. When calculating DM-TAM formation rates, measured data were adjusted for background. Beyond this observation, no further metabolite signals were detected in either the cellular or the microsomal system after incubating TAM-blanks without cellular components, indicating spontaneous decay. However, both experimental systems showed a significant decrease in TAM after incubation, with 40–50% less detectable than at the start. Surprisingly, this reduction was not observed in samples with CHO cells or microsomes, where TAM concentration remained unchanged. In the microsomal system, samples without CHO-based MFs, such as TAM-blanks and HLMs, consistently showed lower TAM concentrations post-incubation (Figure 6A), regardless of NADPH presence. This reduction is unlikely to be due to metabolization. Although HLMs were pre-diluted in the same buffer as CHO MFs for comparison, unknown components in the respective microsomes may be responsible for this effect. Whether this is associated with a possible change in the properties of the substrate cannot be answered at this point. We believe that adsorption effects on the surfaces of pipettes and culture vessels are the primary cause of TAM loss in solution. Preliminary experiments prompted a literature search about this phenomenon prior to the TAM metabolization with mono-CYP CHO models, particularly regarding small hydrophobic molecules like TAM. The observed adsorption phenomenon appears to primarily affect hydrophobic metabolites 4-OH-TAM, 2-methyl derivatives, and TAM itself [56,57]. The effect largely depends on the solvent, the chemical and morphological properties of the contact surface, and substance concentration. The greatest adsorption occurs with the most hydrophobic substances in low concentrations, culture vessels made of plastic (i.e., polystyrene, polycarbonate, polypropylene), and solutions without protein content (e.g., serum). Serum has a stabilizing effect, as TAM can bind well due to its multiple phenyl groups and is thus kept in solution [58,59,60]. Whether this binding is irreversible, and thus, as with adsorption to a surface, removes the substrate from metabolization depends on the protein. Various authors recommend avoiding plastics in TAM studies and using glass instead, emphasizing the importance of determining the actual concentration of the active substance in cell culture experiments to avoid misleading conclusions based on overestimated values [56].

These findings prompted us to conduct TAM metabolization in glass containers and minimize contact with polymer surfaces (pipette tips, serological pipettes, and culture vessels) when handling aqueous TAM solutions. However, the transition to glass vessels did not completely eliminate the loss of TAM in the TAM-blanks. In our opinion, this can be explained by the in vitro models we employed in correlation with findings from the literature already mentioned. The metabolizations were conducted with suspension cells as well as microsomes without the addition of serum proteins, so that the stabilizing proteinogenic effects described in the literature, such as the binding of TAM to proteins and/or the non-specific protein binding on surfaces (e.g., due to the Vroman effect), are only partially effective at reducing adsorption in our models. We consider decisive effects on the quantification of TAM due to solubility effects in the in vitro systems to be unlikely, as precipitation of TAM was not observed in preliminary studies with KHB buffer, cell culture medium (ProCHO5), or in KPO_4_ or during the metabolization experiments. Furthermore, sampling for HPLC quantification was always conducted from homogeneous solutions, which were mixed 1:1 in acetonitrile prior to further processing, ensuring that potentially precipitated TAM at the microscopic level was re-dissolved.

The aforementioned insights from the literature partially explain our observation that TAM significantly decreased after 4 h in TAM-blanks but remained unchanged in CHO cells. Although the KHB buffer used for metabolization with CHO cells lacks proteins or serum, the high cell density of 1.2 × 10^7^ cells/mL likely reduced adsorption to the glass walls. Since this is not a 2D culture, cells were harvested during supernatant collection, and proteins were precipitated with acetonitrile, likely releasing cellular-bound TAM and metabolites for detection. However, this does not fully explain the observations in the microsomal model. Samples with HLMs or CHO MFs had identical compositions, yet significant differences in TAM concentrations were observed after incubation, suggesting unknown factors affecting TAM levels in solution. To clarify the mechanisms, it is essential to identify potential differences in the respective microsomes, such as their production, before conducting further metabolization studies. The impact of reduced TAM availability on microsomal metabolic performance is unclear, but a TAM concentration of ~50 µM was consistently present across all samples, showing no significant decrease over 6 h of metabolization. Thus, we believe a depletive influence on the overall reaction is unlikely.

The metabolization of TAM using the mono-CYP2D6 and mono-CYP3A4 CHO model in both single and combination approach yielded similar results in both in vitro test systems (cellular/microsomal). The metabolite spectra and ratios of detected metabolites were very similar, demonstrating how CYPs interact with TAM to form metabolites. Mono-CYP2D6 CHO was the only model that synthesized detectable amounts of both TAM metabolites, 4-OH-TAM and DM-TAM, as well as the main active metabolite, endoxifen, in both in vitro systems. Mono-CYP3A4 CHO produced significant amounts only of DM-TAM, which was higher than that from mono-CYP2D6 CHO in the cellular system and roughly equivalent in the microsomal system. 4-OH-TAM and endoxifen were not generated by mono-CYP3A4 or were produced only in trace amounts. In addition to the mentioned TAM metabolites, small amounts of other minor metabolites were detected in both in vitro systems, likely originating from 4-OH-TAM and DM-TAM, with clearer detection in the microsomal system. For instance, two weak but consistently detectable metabolite peaks appeared near the retention time of 4-OH-TAM, observed only in microsomal conversions with mono-CYP2D6 and co-CYP2D6/-3A4. These signals were too weak for further characterization but may represent (E)- or (Z)-variants of 4-OH-TAM and 4′-OH-DM-TAM. The minor metabolite DDM-TAM was clearly detected in all but one mono-CYP CHO model, while it was barely detectable in mono-CYP3A4 MF compared with cell experiments.

Interestingly, in the cellular system, more DM-TAM was formed in mono-CYP3A4 than in mono-CYP2D6, but significantly less DDM-TAM was produced. This suggests that physiologically, CYP3A5 may be more involved in DDM-TAM formation. Importantly, our results indicate that CYP2D6 is capable of further demethylation of DM-TAM. If a different mechanism in the CHO cells, rather than recombinant CYP2D6, were responsible, this should have been evident in mono-CYP3A4. The reduced detection of DM-TAM in the microsomal system of mono-CYP3A4 CHO, along with relatively high standard deviations in other metabolites from mono-CYP CHO microsomes, suggests fluctuations in the quality of isolated microsomal fractions. This issue will be addressed in the future with further optimization of the MF isolation protocol. Nonetheless, our findings indicate that the microsomal mono-CYP CHO in vitro test system has greater sensitivity for detecting low-level metabolites compared with intact cells. To our knowledge, the role of CYP2D6 in DDM-TAM formation has not been established. All sources known to us indicate the formation of this side metabolite exclusively by CYP3A4 and -3A5 [29,30,45].

The microsomal in vitro test platform is specific, sensitive, and effective. A comparison of mono-CYP CHO microsomes revealed faster and generally higher concentrations of TAM metabolites—4-OH-TAM, DM-TAM, DDM-TAM, and endoxifen — compared with HLMs. Enzyme activity in mono-CYP CHO MFs approached zero after 2 h, indicating that CYPs in these microsomes convert TAM more efficiently or the holoprotein content is higher than those in HLMs, which showed declining enzyme activity over time. After 6 h, its concentration was comparable only to that of mono-CYP CHO in the formation of DM-TAM. A more detailed assessment is currently not possible, as we have not yet developed a protocol for the quantification of CYP holoenzymes using CO difference spectroscopy in volumes ≤ 50 µL.

The reasons why the TAM metabolism in mono-CYP CHO microsomes ceased after just 2 h also remain unclear. The substrate was consistently present in sufficient amounts, making this an unlikely cause. The same applies to the availability of reaction equivalents such as NADPH, as a regeneration system was employed and the reaction mixtures were supplemented with double the concentration recommended by the manufacturer. It is conceivable that the microsomal CYP enzymes lost their activity over time, thereby preventing further metabolism, or that one of the formed intermediates inhibits further TAM metabolism. Given the complex biotransformation of TAM (especially CYP-dependent), where a CYP enzyme can be involved in the formation of multiple TAM metabolites, effects such as feedback inhibition are plausible. Therefore, investigating these intriguing observations will be the subject of future studies.

Regarding the use of multi-CYP models, we found no evidence of synergistic effects in the metabolization of TAM in either co-CYP2D6/-3A4 cells or mixed microsomal preparations. Neither co-culture nor equivalent mixtures of microsomes from both mono-CYP CHO models showed significantly altered formation of TAM metabolites. The metabolite spectrum corresponded to that of CYP2D6, with metabolite amounts consistently falling in the middle between the results of the individual mono-CYP CHO models.

The results clearly showed that small amounts of DM-TAM were detected on both the cellular and microsomal level against the background detected in TAM-blanks in the parental CHO. This indicates that there was CYP activity. It is highly unlikely that CYP-independent formation of DM-TAM occurred, as no TAM metabolites were detected in CHO MFs without NADPH supplementation. However, the activity does not seem to originate from any of the CYPs we have investigated so far, as no enzyme activity could be detected. GloCYP activity assays were performed for CYP1A2, -2B6, -2D6, -3A4, and -2C19 (the latter is unpublished) as well as previous cellular metabolization studies with prototypical substrates such as phenacetin or testosterone for CYP1A2 or CYP3A4. These showed no background CYP activity of parental CHO cells [19]. Other CYPs involved in the formation of DM-TAM are CYP1A1, -2C9, and 3A5 [29,30,45]. To assess the mono-CYP CHO platform, these and possibly other CYP activities should be successively screened in the CHO cell model.

## 4. Materials and Methods

### 4.1. Cell Culture and Generation of Recombinant CPR/CYP2D6 (Mono-CYP2D6) Expressing CHO Cells

Routine cultivation of suspension-adapted Chinese Hamster Ovary cells (originally CHO-K1, CCL-61 clone; ATCC, MNZ, VA, USA) was performed in ProCHO5 medium (Lonza Group AG, Basel, Switzerland) supplemented with 6 mM L-alanyl-L-glutamine (Biowest, Nuaillé, France). For cell expansion, suspensions were cultured at 120 rpm on an orbital shaker in 125 mL baffled flasks (30 mL culture, Corning, NY, USA) at 37 °C and 5% CO_2_. For some characterizations, such as the initial determination of CYP2D6 enzyme activity in intact mono-CYP CHO 2D6 clones recombinantly expressing human CPR and CYP2D6, cultivation in adherence was necessary. For this purpose, the culture medium was supplemented with 2% FBS (Bio&Sell GmbH, Feucht, Germany), and the cells were cultivated in plasma-treated TCP vessels (STARLAB GmbH, Hamburg, Germany).

In preliminary studies with CHO-CPR/CYP (mono-CYP CHO) suspension models, several supplementation studies with heme precursors were performed to increase CYP holoenzyme content and, thereby, overall CYP enzyme activity. Based on these experiments, a pre-treatment protocol for CHO cells prior to metabolization studies was established, which was also used for the metabolization of tamoxifen (TAM) shown here. It includes the pre-treatment with 2% DMSO in the culture medium for 24 h prior to metabolization with suspension cells or isolation of MFs for later microsomal-based TAM metabolization.

The generation, selection, and isolation of mono-CYP2D6 CHO single clones was performed according to the protocol previously described for the generation of corresponding mono-CYP3A4, -1A2, and -2B6 CHO cell lines with recombinant human CPR and specific CYP activity [19]. The coding sequence for human CYP2D6-cDNA used for lentiviral gene transfer was NM_001025161 (NCBI ref. seq.), and the lentiviral expression vector pLenti4/V5-DEST (Thermo Fisher Scientific Inc., Waltham, MA, USA) additionally contained a Zeocin resistance gene for selection of successfully transfected clones. Recombinant lentivirus was used to infect the previously generated recombinant human CPR-expressing CHO-CPR C12 clone to generate recombinant human CPR and CYP2D6-expressing mono-CYP2D6 CHO cell lines. Selection of the clones was performed by cultivation with 3 µg/mL Blasticidin (resistance of CHO-CPR C12; AppliChem GmbH, Darmstadt, Germany) and 300 µg/mL Zeocin (Thermo Fisher Scientific Inc.).

### 4.2. Preparation of Microsomal Fractions from CHO Models

Microsomal fractions (MFs) were extracted from 3 × 10^7^ freshly harvested suspension cells (CHO and recombinant CPR/CYP-expressing clones) after pre-treatment with DMSO described above. Briefly, the cells were centrifuged for 5 min with 125× *g* at 4 °C, washed with cold PBS (Biowest), and re-suspended in 500 µL homogenization buffer of the used Microsome Isolation Kit (BioVision Inc., Milpitas, CA, USA). Cell lysis was induced by treating the cells in an ice bath with 3 × 10 s ultrasonic pulses at 40% amplitude intermitted by 20 s breaks to prevent sample overheating using a Sonopuls ultrasonic homogenizer equipped with an UW3100 ultrasonic probe (BANDELIN electronic GmbH & Co. KG, Berlin, Germany). Subsequently, extraction of microsomes was performed according to the manufacturer’s protocol. Finally, MFs were eluted in 150 µL cold storage buffer, aliquoted, and stored at −80 °C until use. For protein quantification, an aliquot of each isolated MF was quantified by Pierce™ BCA Protein Assay Kit (Thermo Fisher Scientific Inc.) against a BSA standard according to the manufacturer’s instructions.

### 4.3. CPR and CYP Protein Expression Analysis by Immunodetection

Proteins for immune detection of recombinant expressed CPR and CYP2D6 in mono-CYP2D6 CHO clones by Western Blot analysis were prepared as previously described [19]. Immune detection of CPR was achieved by using a polyclonal rabbit-anti-CPR IgG as primary antibody (1 mg/mL, Abcam, Cambridge, UK) diluted 1:1000 in 2% BSA. CYP2D6 detection occurred by using a monoclonal rabbit-anti-CYP2D6 IgG as primary antibody diluted 1:1000 in 2% BSA (EPR17868; 1.5 mg/mL; Abcam). As loading control, GAPDH was detected using a monoclonal mouse-anti-GAPDH IgG (0.5 mg/mL, antibodies-online GmbH, Aachen, Germany) as primary antibody in 1:5000 dilution. Primary antibody binding was performed at 4 °C overnight. Incubation of the blots with peroxidase-conjugated secondary antibodies goat-anti-mouse IgG or goat-anti-rabbit IgG (both 1:2000 in 2% BSA in PBS, Merck KGaA, Darmstadt, Germany) at ambient temperature for 1 h facilitated protein detection by enhanced chemiluminescence reaction using the Amersham Prime Western Blotting Detection Reagent (GE Healthcare, Chicago, IL, USA) in combination with a Biostep Celvin^®^ S 420 chemiluminescence imaging system (Biostep GmbH, Burkhardtsdorf, Germany). Recombinant human CYP2D6 expressing HepG2-CYP2D6 cells, as well as parental CHO cells and the CHO-CPR C12 clone, served as reference.

To validate the successful isolation of microsomes from mono-CYP2D6 C18 and mono-CYP3A4 C1 CHO cells, both CPR and the respective CYP enzyme were detected at the protein level from isolated MFs. The parallel detection of voltage-dependent anion channel 1 (VDAC-1) served to assess the quality of MFs. Immune detection was performed analogously to that for cellular CPR/CYP expression. Briefly, 20 µg protein of each MF was applied to the gel and separated by SDS-PAGE. For the immune detection of CYP2D6, the previously mentioned antibodies were used in comparable concentrations. CYP3A4 was detected by using a monoclonal mouse-anti-CYP3A4 IgG1 as primary antibody (clone: 3H8; Thermo Fisher Scientific Inc.) diluted 1:1000 in 2% BSA. VDAC-1 detection was realized by using a polyclonal rabbit-anti-VDAC-1 IgG (Abcam) as primary antibody diluted 1:2000 in 2% BSA. Peroxidase conjugated secondary antibodies were goat-anti-mouse IgG or goat-anti-rabbit IgG (both 1:2000 in 2% BSA in PBS, Merck KGaA). MFs from parental CHO cells and commercially available human liver microsomes (HLMs, pool from 50 donors, Thermo Fisher Scientific Inc.) were used as references.

### 4.4. Selection of the Most Promising Mono-CYP2D6 CHO Clone by CYP Activity Determination

To select the mono-CYP2D6 clone with the highest CYP2D6 activity, individually isolated clones were analyzed by P450-Glo™ CYP2D6 assay (Promega, Madison, WI, USA). Data were collected with n = 3 for cells and n = 12 from 2 independent experiments for isolated MFs. Briefly, for initial assessment of the clones’ CYP2D6 activity and to narrow down to the three best performing cell lines, adherent and confluent grown mono-CYP2D6 clones were incubated with 50 µM of the luminogenic CYP2D6 substrate luciferin-ME EGE diluted in culture medium at 37 °C, 5% CO_2_ for 60 min. After incubation, 25 μL of the supernatant was transferred into a white-walled 96-well plate (SARSTEDT AG & Co. KG, Nümbrecht, Germany), and an equal volume of luciferin detection reagent was added, followed by 20 min incubation at ambient temperature. Subsequently, luminescence intensity was measured with a FLUOstar Omega microplate reader (Software version: 3.00 R2, BMG LABTECH GmbH, Ortenberg, Germany), followed by data analysis by MARS Data Analysis Software (Version: 2.41). In addition, the cells and the 25 μL substrate solution remaining in the initial plate were mixed with 25 μL ATP reagent of the CellTiter-Glo^®^ 2.0 assay (Promega) and incubated for 10 min in the dark. ATP levels were detected by measuring luminescence to allow normalization to the cell number. As references, HepG2-CYP2D6, parental CHO, and CHO-CPR C12 cells were included.

For the final selection of the most active mono-CYP2D6 CHO clone, MFs of C1, C9, and C18 as well as the parental cell line CHO-CPR C12 (background control) were isolated and quantified according to the protocol described in Section 4.2. A protocol of the P450-Glo™ CYP2D6 assay adapted for microsomes was subsequently used to quantitatively determine CYP2D6 activity against a luciferin standard (Beetle Luciferin; Promega) in the range between 0.0016 and 4 µM. HLMs served as physiological reference. Briefly, the activity was determined using 1 µg microsomal protein in 50 µL potassium phosphate buffer (KPO_4_, 0.1 mM, pH 7.4) per well in 96-well format (white-walled plate). Metabolization was performed with 30 µM luciferin-ME EGE for 30 min at 37 °C. To provide reduction equivalents during metabolization, the NADPH Regeneration System (Promega) was used according to the manufacturer’s recommendation. After incubation, 50 µL luciferin detection reagent was added, followed by 20 min incubation at ambient temperature. Subsequently, luminescence intensity was measured as described above. All samples were measured with n = 6 wells from two independent preparations.

### 4.5. Metabolization of Tamoxifen by Mono-CYP CHO Cells and Microsomes

Metabolization of TAM with intact cells was performed in Krebs Hanseleit buffer (KHB, 9.6 g/L KHB, 25 mM NaHCO_3_, 25 mM HEPES, 2 mM CaCl_2_, pH 7.4, all from Merck KGaA) with 3 × 10^7^ cells in 2.5 mL suspension culture (1.2 × 10^7^ cells/mL). Mono-CYP2D6 C18 and mono-CYP3A4 C1 (previously published [19]), both in co-culture (equal mixture), were treated with 50 µM TAM (50 mM stock solution prepared in pure EtOH and diluted in KHB) compared with parental CHO cells for 4 h (n = 4 independent metabolization, quantified twice by HPLC). During metabolization, cell suspensions were incubated under culture conditions and orbital shaking at 150 rpm in 10 mL Duran laboratory bottles (VWR, Radnor, PA, USA). To monitor possible TAM adsorption to the culture vessel during metabolization, samples containing TAM in KHB but without cells were analyzed in parallel. To calculate the enzyme activity for TAM metabolite generation, 100 µL aliquots of the cell suspensions were secured after the metabolization reaction. The cells were pelleted by centrifugation for 5 min with 125× *g* at 4 °C, and the protein concentration was subsequently determined using the Pierce™ BCA Protein Assay Kit.

Metabolization of TAM with pre-isolated MFs was performed with 200 µL preparations in 2 mL amber glass bottles (Agilent Technologies Inc., Santa Clara, CA, USA). An amount of 0.1 mM TAM (50 mM stock freshly dissolved in pure EtOH and pre-diluted in water) was metabolized with 100 µg microsomal protein in KPO_4_ buffer and twice the amount of NADPH Regeneration System suggested by the manufacturer at 37 °C for 2, 4, and 6 h, respectively (n = 3 from three independent experiments and two microsome preparations). Corresponding to the cell metabolization, MFs of mono-CYP2D6 C18, mono-CYP3A4 C1, and equal mixtures of both (co-CYP2D6/-3A4 MF) were investigated compared with parental CHO MFs and HLMs as physiological reference. TAM without microsomes, as well as parental CHO MFs and HLMs without adding the NADPH Regeneration System, served as additional controls checking for substrate adsorption and CYP-independent side reactions.

TAM metabolite profiling and quantification was based on HPLC analysis of cell culture supernatants and MFs following metabolization. After collection, samples were mixed with an equal volume of ice-cold acetonitrile (VWR, Radnor, PA, USA) spiked with 100 µM Phenacetin (Merck KGaA) as internal standard. After addition, the samples were vortexed, incubated on ice for 5 min, and centrifuged at 16,000× *g* at 4 °C for 15 min. Supernatants obtained were analyzed by HPLC. The setup consisted of a SCL-40 (SHIMADZU, Kyoto, Japan) with a ZORBAX SB-C18 column (Agilent Technologies Inc.). For metabolite separation, 20 µL sample was separated in a gradient of KH_2_PO_4_ buffer (10 mM, pH 3, mobile phase A) and acetonitrile (mobile phase B) at a flow rate of 1 mL/min and 50 °C column temperature. Metabolites were detected based on elution time using DAD (SPD-M40) at 280 nm. The gradient was as follows: 0–3 min 30% B, 20.5 min 55% B, 22.5 min 55% B, 23 min 95% B, 26 min 95% B, 26.5 min 20% B, 29 min 20% B, and 30 min 30% B. Quantification of detected metabolites was performed using standard solutions of commercially available references (TAM; *N*-desmethyltamoxifen (DM-TAM), 4-hydroxytamoxifen (4-OH-TAM), and endoxifen, all from Merck KGaA) in the range of 10–100 µM. Stock solutions of substrates and references were dissolved in pure EtOH and further diluted in the respectively used solvent (KHB or KPO_4_ buffer).

### 4.6. Statistical Analysis

Depending on the data, one- or two-way ANOVA with Tukey’s multiple comparison test was used to probe for significant differences between groups using Prism 8 (GraphPad Software, Version: 8.2.1, San Diego, CA, USA). Statistical significance was assumed for *p* < 0.05 for all statistical data.

## 5. Conclusions

Our mono-CYP CHO platform enabled us to reproduce scientifically accepted knowledge on CYP-dominated TAM metabolization and revealed new aspects regarding the dependence of TAM metabolism on specific CYP enzymes.

CYP2D6 is a key enzyme in the synthesis of endoxifen, which is generally assumed to be the main therapeutically relevant metabolite. CYP3A4 produced appreciable amounts of DM-TAM, but not 4-OH-TAM and endoxifen. In addition, our results definitively indicated that CYP2D6 is also involved in the formation of the minor metabolite DDM-TAM, which to the best of our knowledge has not been confirmed or addressed in the past. As we stated, we were able to show that mono-CYP CHO models in the form of intact cells as well as microsomes provide comparable and reproducible results in the CYP-dependent metabolization of TAM.

The metabolite spectra of both mono-CYP models illustrate their CYP specificity and sensitivity, especially in microsomal in vitro models, where these were significantly higher compared with HLMs. CYP-specific major as well as minor metabolites could be detected in shorter conversion times and often at significantly higher concentrations. We also detected signals for other TAM side metabolites formed by CYPs. However, we were unable to elucidate the identity of these metabolites in this study. These were detected only in the mono-CYP CHO models and not in HLMs. Nevertheless, we observed no significant synergistic effects between CYP2D6 and CYP3A4 in either the cellular or microsomal in vitro system. In the course of our experiments, it also became clear that the adsorption of TAM on culture vessels could be a significant influencing factor, with considerable proportions of the substrate disappearing from the solution after a short time. This raises the question of the extent to which the drug’s reduced effective concentration influences a reaction and whether or to what extent the metabolites formed are also affected by this phenomenon. These are aspects that should be investigated in more detail to assess the influence of adsorption for future metabolization studies.

## Figures and Tables

**Figure 1 ijms-26-03992-f001:**
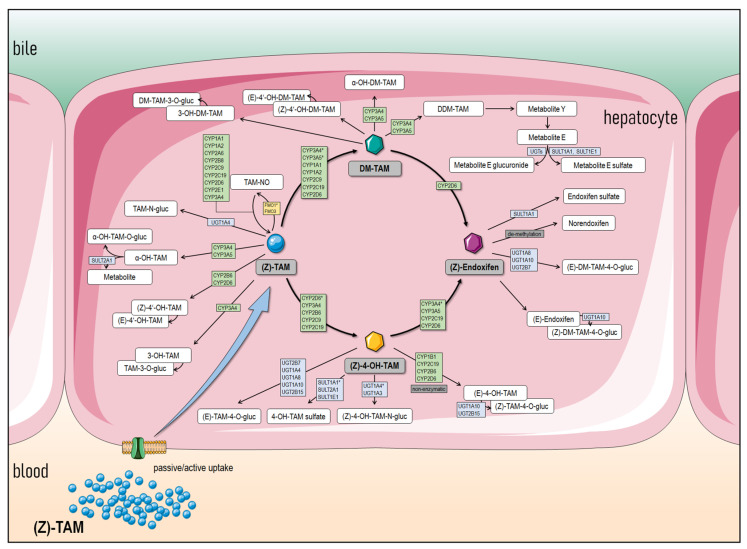
TAM metabolization pathway. Combined metabolic pathway prediction of TAM in the human liver adapted from different sources [29,30,45] (* main responsible enzyme for metabolization. Illustration modified by Servier Medical Art—License CC BY 3.0; https://smart.servier.com (accessed on 17 April 2025)).

**Figure 2 ijms-26-03992-f002:**
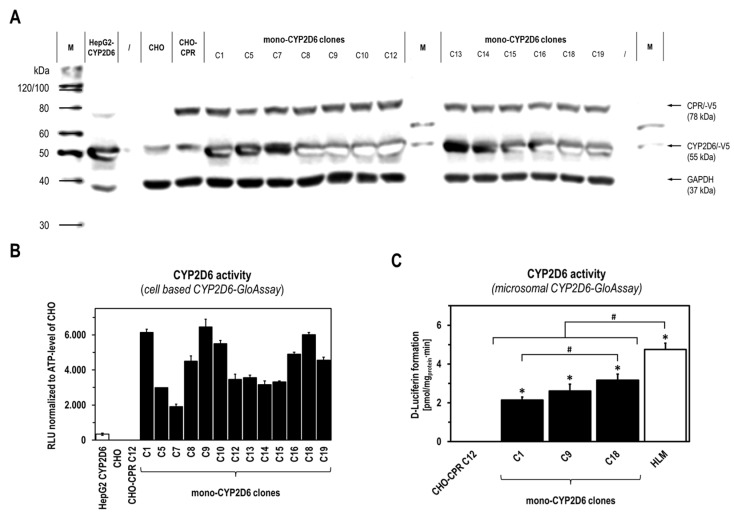
Recombinant CPR/CYP2D6 expression and CYP2D6 activity in selected mono-CYP2D6 clones. Recombinant expression of human CPR and CYP2D6 as well as CYP2D6 activity was initially analyzed for isolated mono-CYP2D6 clones at the protein level by Western Blotting (**A**) and cell-based GloCYP2D6 activity assay (**B**), with HepG2-CYP2D6, CHO-CPR C12, and parental CHO cells as references. CYP2D6 activity of MF from selected mono-CYP2D6 clones C1, C9, and C18 compared with CHO-CPR C12 and HLMs using microsomal-based GloCYP2D6 activity assay (**C**) (HLMs = human liver microsomes; CYP2D6 activity data with n = 3 for cells and n = 6 for microsomes presented as mean ± standard deviation; one-way ANOVA with Tukey’s multiple comparison test was used to probe for significant differences between clones; *p* < 0.05 with * compared with CHO-CPR C12; ^#^ compared between mono-CYP2D6 clones and HLMs).

**Figure 3 ijms-26-03992-f003:**
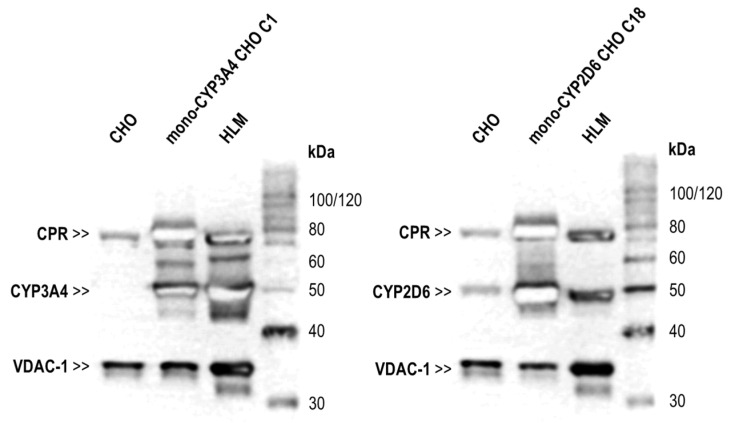
CPR and CYP protein levels in mono-CYP CHO microsomal fractions (MFs). Analysis of CPR, CYP, and VDAC-1 protein levels in isolated MFs by immune detection from both mono-CYP CHO models used for TAM metabolization compared with parental CHO MF and commercial HLMs as references. Both mono-CYP CHO MFs showed high CPR and CYP (CYP2D6 or -3A4, respectively) protein levels compared with parental CHO MFs. CYP2D6 but not CYP3A4 was also detectable in parental CHO MFs. In HLMs, both CYP and CPR amounts in between the mono-CYP CHO and parental CHO MFs were visible. Further, HLMs showed higher VDAC-1 protein levels compared with microsomes from CHO models.

**Figure 4 ijms-26-03992-f004:**
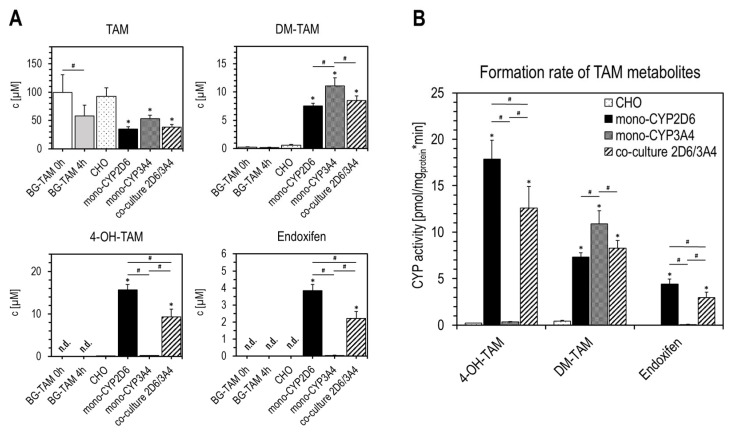
Generation and formation rate of TAM main metabolites by mono-CYP CHO cells. Concentration of the main metabolites DM-TAM, 4-OH-TAM, and endoxifen as well as the substrate TAM in culture supernatants after 4 h incubation (**A**). Formation rate of TAM metabolites in the mono-CYP CHO models compared with CHO parent cells (**B**) (data presented as mean ± standard deviation; one-way ANOVA with Tukey’s multiple comparison test was used to probe for significant differences between groups; *p* < 0.05 with * compared with parental CHO cells; # compared between samples; n = 4).

**Figure 5 ijms-26-03992-f005:**
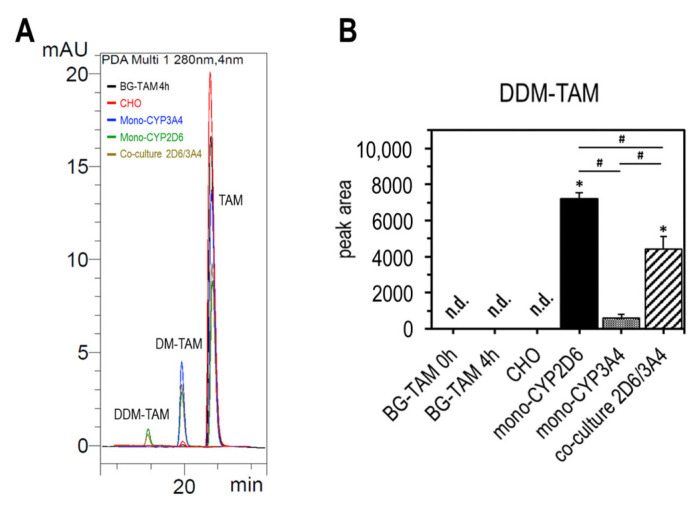
Formation of DDM-TAM by mono-CYP2D6 CHO. Comparison of the formation of DM-TAM and DDM-TAM in CHO cell models after 4 h metabolization detected by HPLC-DAD. Overlay of HPLC spectra in the retention region of TAM and its demethylated metabolites (**A**) and peak areas of the detected signal for DDM-TAM (**B**) (data presented as mean ± standard deviation; one-way ANOVA with Tukey’s multiple comparison test was used to probe for significant differences between groups; *p* < 0.05 with * compared with parental CHO cells; # compared between samples; n = 4).

**Figure 6 ijms-26-03992-f006:**
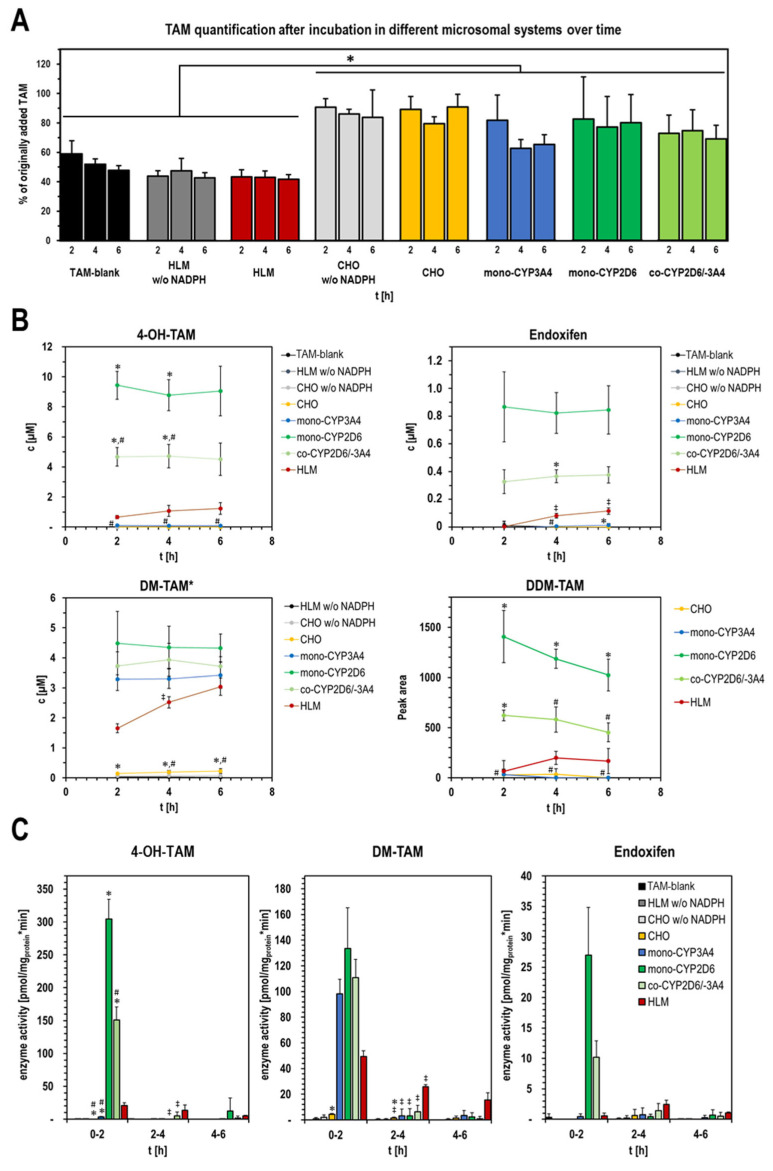
CYP specific metabolization pattern of TAM by mono-CYP CHO microsomal fractions. Quantification of TAM revealed differences in the dynamics of TAM availability in the used microsomal models during metabolization (**A**). Concentration/peak area (**B**) and metabolite-specific enzyme activity (**C**) of TAM metabolites formed during metabolization for up to 6 h with MFs of mono-CYP2D6, mono-CYP3A4, and co-CYP2D6/-3A4 compared with parental CHO MF and HLMs as physiological reference. Representation of detected intensities for the suspected side metabolite DDM-TAM as peak area over time (* background corrected data of DM-TAM; data presented as mean ± standard deviation; two-way ANOVA with Tukey’s multiple comparison test was used to probe for significant differences between groups; except of (**A**), *p* < 0.05 with * CHO MF models compared with HLMs; # compared with mono-CYP2D6 MF and ^‡^ compared with previous time point of a sample; n = 3 independent metabolizations from two microsomal isolations).

## Data Availability

All data are available on reasonable request from the corresponding author.

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
