# Peer review of "Mono-CYP CHO Model: A Recombinant Chinese Hamster Ovary Cell Platform for Investigating CYP-Specific Tamoxifen Metabolism"

_ijms, 2025, doi:10.3390/ijms26093992_

Round 1

Reviewer 1 Report

Comments and Suggestions for Authors

Authors developed the CHO cell lines expressing human CYP2D6 with CPR. Together with the previously developed CHO expressing CYP3A4 CPR, they evaluated TAM metabolization using cells and microsome fractions. The results are interesting and worth for publication in IJMS. However, I have some concerns shown below.

  1. It is difficult to understand the descriptions in lines 612-626. The enzymatic activity must be normalized by protein quantity and not by cell numbers. The calculation appears to be incorrect. Cells need to produce 100 microgram of liver microsomal protein is 2.4 x 105cells not 2.4 x 108 cells. (50 mg/1.2 x 108cells = 100 microgram/2.4 x 105 cells), therefore 25 times fewer liver cells in the form of HLM were used for the assay that mono-CYP CHO cells. In my understanding, CHO cells and human hepatocytes are completely different in size and development of organelles, therefore they are not comparable.
  2. As shown in figure 6, TAM metabolization in mono-CYP CHO MF models stopped following 2-hour incubation although most of TAM in the system remained. Why? Did one of the intermediates inhibit metabolization?

Author Response

Dear Reviewer,

Thank you for providing us with the reviewers’ comments on our manuscript “Mono-CYP CHO model: A recombinant Chinese Hamster Ovary cell platform for investigating CYP-specific Tamoxifen metabolism”, which helped us improve our work. In the following, we answer the reviewer 1 comments point-by-point and explain the changes incorporated in our revised manuscript. The reviewers’ comments are in italic, the related changes in our manuscript in light blue and our response in green.

Please do not hesitate to contact me or my co-authors in case you need further information or have questions.

With best regards,

Christian Schulz

Reviewers' Comments:

Reviewer’s comment #1-1:

Authors developed the CHO cell lines expressing human CYP2D6 with CPR. Together with the previously developed CHO expressing CYP3A4 CPR, they evaluated TAM metabolization using cells and microsome fractions. The results are interesting and worth for publication in IJMS. However, I have some concerns shown below.

  1. It is difficult to understand the descriptions in lines 612-626. The enzymatic activity must be normalized by protein quantity and not by cell numbers. The calculation appears to be incorrect. Cells need to produce 100 microgram of liver microsomal protein is 2.4 x 105cells not 2.4 x 108 (50 mg/1.2 x 108cells = 100 microgram/2.4 x 105 cells), therefore 25 times fewer liver cells in the form of HLM were used for the assay that mono-CYP CHO cells. In my understanding, CHO cells and human hepatocytes are completely different in size and development of organelles, therefore they are not comparable.

Our response to comment reviewer #1-1:

We thank the reviewer for the assessment of our manuscript and for pointing out, that the enzymatic activity for the formation of a specific metabolite must be normalized to the amount of protein. The reviewer is right about this and we described it in 2.4 and Fig. 6C for the metabolization of TAM by HLM and mono-CYP CHO. In the comparison described in lines 612 - 626, we were concerned with comparing the cell quantities required to provide a certain microsomal protein quantity from HLM or mono-CYP CHO as an additional information about the in-vitro model. Following the reviewer's comment, we realized that our wording in the context of the manuscript did not make this clear to the reader. The reviewer is also correct that the calculation of the respective cell numbers was incorrect. Nevertheless, we agree with the reviewer's opinion that a direct comparison of required cell numbers is not possible due to the large differences between human hepatocytes and CHO cells.

In response to the reviewer's helpful comments, we have therefore supplemented the text in and removed the calculation in lines 612 - 626 from the manuscript.

Adjustments in the manuscript:

line: 627 - 632

“… that CYPs in these microsomes convert TAM more efficiently or the holoprotein content is higher than those in HLM, which showed declining enzyme activity over time. After 6 h, its concentration was only comparable to that of mono-CYP CHO in the formation of DM-TAM. A more detailed assessment is currently not possible, as we have not yet developed a protocol for the quantification of CYP holoenzymes using CO difference spectroscopy in volumes ≤50 µL. ...”

Reviewer’s comment #1-2:

  1. As shown in figure 6, TAM metabolization in mono-CYP CHO MF models stopped following 2-hour incubation although most of TAM in the system remained. Why? Did one of the intermediates inhibit metabolization?

Our response to comment reviewer #1-2:

We thank the reviewer for his comment and would like to comment on it.

We are not yet sure what caused the TAM metabolism in the mono-CYP CHO MF models to stop after only 2 h. The substrate was present in sufficient quantities, so we do not believe that this was a cause. One possibility would be that reaction equivalents such as NADPH were depleted. However, this is unlikely, as a regeneration system (NADP+, G6P and G6P-dehydrogenase) was used and the reaction preparations were supplemented with twice the concentration suggested by the manufacturer (previously titrated out by us in preliminary studies). In our opinion, other possibilities could be that microsomal CYP enzymes lose their activity over time and thus no further conversion is possible or, as commented by the reviewer, one of the intermediates formed inhibits further TAM metabolization. In view of the complex biotransformation of TAM (especially CYP-dependent), in which one CYP can be involved in the generation of several TAM metabolites, effects such as feedback inhibition are conceivable. For example, such effects on the gene expression of CYP2C19 have been described for clopidogrel, whereby inhibition in the present TAM model would have to act directly on the enzyme.

We thank the reviewer for his comment and input on this aspect. We are already planning follow-up studies to get to the bottom of these observations. We hope that we were able to answer the reviewer's question satisfactorily and have emphasized this aspect more clearly in the manuscript for the readers.

Adjustments in the manuscript:

line: 633 - 643

“… The reasons why the TAM metabolism in mono-CYP CHO microsomes ceased after just 2 h also remain unclear. The substrate was consistently present in sufficient amounts, making this an unlikely cause. The same applies to the availability of reaction equivalents such as NADPH, as a regeneration system was employed and the reaction mixtures were supplemented with double the concentration recommended by the manufacturer. It is conceivable that the microsomal CYP enzymes lost their activity over time, thereby preventing further metabolism, or that one of the formed inter-mediates inhibits further TAM metabolism. Given the complex biotransformation of TAM (especially CYP-dependent), where a CYP enzyme can be involved in the formation of multiple TAM metabolites, effects such as feedback inhibition are plausible. Therefore, investigating these intriguing observations will be the subject of future studies. …”

Reviewer 2 Report

Comments and Suggestions for Authors

In this manuscript, the authors reported the evaluation of the performance of their mono-CYP CHO platform utilizing tamoxifen as a model drug. The metabolism of tamoxifen was studied utilizing mono-CYP2D6 and mono-CYP3A4 clones separately and in combination, which was compared to the human liver microsome system. The mono-CYP CHO platform allows in vitro metabolism investigations at the cellular and microsomal levels, which should be a valuable tool for studying drug metabolism.

The manuscript is in general easy to follow. However, there are a few issues that need to be addressed:

  1. Conflicting information: Figure 2 shows lower activity of mono-CYP2D6 clones 1 and 9 compared to 18 while the text above the figure states “no significant difference in the CYP2D6 band intensity could be detected between the mono-CYP2D6 CHO clones”.
  2. HLM has higher level of CYP3A4 according to Figure 3 (left blot) and the text above the figure. However, the figure caption claims that “Both mono-CYP CHO MF showed high CPR and CYP (CYP2D6 or -3A4, respectively) protein levels”. In addition, why did mono-CYP2D6 clones have lower activity when their protein levels were higher than in HLM?
  3. Regarding the unexpected observation of reduction of tamoxifen concentrations over time in the TAM-blanks: The authors suspected non-specific binding, which was reasonable. However, did the authors repeat the experiment and obtain similar results? After the authors switched to glass containers and minimized contact with polymer surfaces, it seemed the problem didn’t go away completely. Is it possible that this was due to limited solubility of the drug so some of the drug precipitated overtime? Did they conduct solubility experiments in various matrices to ensure that the drug concentrations used in the metabolism studies didn’t exceed the solubility limit?
  4. Pre-treating CHO cells with DMSO: Isn’t 2% DMSO too toxic to the cells?

Major revision is recommended prior to reconsideration of the manuscript for publication.

Author Response

Dear Reviewer,

Thank you for providing us with the reviewers’ comments on our manuscript “Mono-CYP CHO model: A recombinant Chinese Hamster Ovary cell platform for investigating CYP-specific Tamoxifen metabolism”, which helped us improve our work. In the following, we answer the reviewer 2 comments point-by-point and explain the changes incorporated in our revised manuscript. The reviewers’ comments are in italic, the related changes in our manuscript in light blue and our response in green.

Please do not hesitate to contact me or my co-authors in case you need further information or have questions.

With best regards,

Christian Schulz

Reviewers' Comments:

Reviewer’s comment #2-1:

In this manuscript, the authors reported the evaluation of the performance of their mono-CYP CHO platform utilizing tamoxifen as a model drug. The metabolism of tamoxifen was studied utilizing mono-CYP2D6 and mono-CYP3A4 clones separately and in combination, which was compared to the human liver microsome system. The mono-CYP CHO platform allows in vitro metabolism investigations at the cellular and microsomal levels, which should be a valuable tool for studying drug metabolism.

The manuscript is in general easy to follow. However, there are a few issues that need to be addressed:

  1. Conflicting information: Figure 2 shows lower activity of mono-CYP2D6 clones 1 and 9 compared to 18 while the text above the figure states “no significant difference in the CYP2D6 band intensity could be detected between the mono-CYP2D6 CHO clones”.

Our response to comment reviewer #2-1:

We thank the reviewer for his comment and would like to comment on it.

The authors do not see contradictory information in the description of this aspect. In our opinion, it is not contradictory that different cell clones with comparable CYP2D6 protein expression in immunodetection can exhibit varying CYP2D6 activities. In lines 227-229, reference is made to the immunodetection of recombinant protein expression in the mono-CYP2D6 CHO clones. All cell clones show a comparably band intensity, indicating increased CYP2D6 protein expression, compared to the parental cell line and the CHO-CPR C12 cells. Here, the immunodetection is not sensitive enough to reveal potential differences in protein expression. Furthermore, we believe that increased protein expression does not allow for conclusions about the presence of functional CYP holoprotein. Depending on the primary antibody used or its bound epitope, it is also possible to detect non-functional apoprotein. To select the most suitable mono-CYP2D6 cell clone, we therefore measured CYP2D6 activity at the cellular level (Fig. 2B) and subsequently with isolated microsomes from selected mono-CYP2D6 CHO clones (Fig. 2C) using the GloAssay (Promega), which are highly sensitive. Here, the differences between clones C1, C9, and C18 were not orders of magnitude apart.

However, we may not have clearly articulated the analytical approach for the readers. Therefore, we would like to thank the reviewer for the helpful suggestion and have adjusted the text accordingly.

Adjustments in the manuscript:

line: 232 - 237

“… Since the sensitivity of immunodetection was not sufficient to detect possible differences in CYP2D6 protein expression and no conclusions could be drawn about the presence of functional CYP2D6 enzyme, further characterization of the mono-CYP2D6-CHO clones was performed by determining CYP2D6 activity at the level of intact cells. This analysis revealed a heterogeneous CYP2D6 activity in the clone population analyzed (Fig. 2B). ...”

Reviewer’s comment #2-2:

  1. HLM has higher level of CYP3A4 according to Figure 3 (left blot) and the text above the figure. However, the figure caption claims that “Both mono-CYP CHO MF showed high CPR and CYP (CYP2D6 or -3A4, respectively) protein levels”. In addition, why did mono-CYP2D6 clones have lower activity when their protein levels were higher than in HLM?

Our response to comment reviewer #2-2:

We thank the reviewer for his comment and would like to comment on it.

The reviewer is correct that the description for Figure 3 is not clearly articulated. "High CPR and CYP protein levels" refers to the parental cell line and not to HLM. We have revised the figure legend accordingly for clarity.

We would also like to comment on the reviewer's question regarding why, in our opinion, mono-CYP2D6 clones may exhibit lower activity, despite their CPR and CYP2D6 protein concentrations being higher than in HLM. This aspect has already been discussed in the manuscript, and we would like to direct the reviewer to lines 478 - 483, which we have also included here for better clarity.

Lines 478 - 483: „In comparison, HLM showed about 50% higher CYP enzyme activity. According to the manufacturer, the substrate luciferin-ME EGE is not CYP2D6-specific and is also metabolized into luciferin by CYP1A1, -1A2 and -2B6. Unlike the mono-CYP CHO, HLM possess a broad spectrum of CYP enzymes [4,10,24]. Thus, these additional enzymes likely contribute to the detected activity.”

We hope that our explanation satisfactorily addresses the reviewer's question.

Adjustments in the manuscript:

line: 283

“… protein levels compared to parental CHO MF. CYP2D6 but not CYP3A4 were also detectable in …”

Reviewer’s comment #2-3:

  1. Regarding the unexpected observation of reduction of tamoxifen concentrations over time in the TAM-blanks: The authors suspected non-specific binding, which was reasonable. However, did the authors repeat the experiment and obtain similar results? After the authors switched to glass containers and minimized contact with polymer surfaces, it seemed the problem didn’t go away completely. Is it possible that this was due to limited solubility of the drug so some of the drug precipitated overtime? Did they conduct solubility experiments in various matrices to ensure that the drug concentrations used in the metabolism studies didn’t exceed the solubility limit?

Our response to comment reviewer #2-3:

We thank the reviewer for his comment and we will strive to answer his questions to the best of our ability.

The reduction of TAM concentration in TAM blanks has been independently observed several times in cellular (n = 4) and microsomal (n = 3) metabolizations. The results were consistently comparable, and the corresponding data are presented in the manuscript.

The transition to glass vessels did not, as the reviewer correctly noted, completely eliminate the loss of TAM in the TAM blanks. In our opinion, this can be explained by the in-vitro models we employed in correlation with findings from the literature, which we discuss in the manuscript (lines 543 - 558). Since pipetting steps using polymer pipette tips cannot be entirely avoided, losses still occur in this context. Furthermore, the metabolizations were conducted with suspension cells as well as microsomes without the addition of serum proteins, so that stabilizing, proteinogenic effects described in the literature, such as the binding of TAM to proteins and/or the non-specific occupation of surfaces by proteins (e.g., due to the Vroman effect), are only partially effective in reducing adsorption in our models. However, particularly high cell quantities used in the reaction setup have a similar effect, resulting in lower losses when using parental CHO cells.

Visible precipitation of TAM (e.g., through turbidity of the reaction solution) was not observed in preliminary studies with KHB buffer, cell culture medium (ProCHO5), and in KPO4 (microsomal system), as well as during the metabolization experiments. Furthermore, the amount of TAM used was repeatedly recovered via HPLC in metabolizations with parental CHO cells and microsomes, which, in our opinion, renders a significant influence of solubility effects on HPLC quantification improbable. Sampling for HPLC was always conducted from well-mixed homogeneous solutions, and supernatants were mixed 1:1 in the organic solvent acetonitrile prior to further processing, ensuring that potentially precipitated TAM at the microscopic level was re-dissolved.

We hope we were able to answer the reviewer's questions satisfactorily and have also discussed this aspect more clearly in the manuscript.

Adjustments in the manuscript:

line: 561 - 575

“… vessels) when handling aqueous TAM solutions. However, the transition to glass vessels did not completely eliminate the loss of TAM in the TAM-blanks. In our opinion, this can be explained by the in-vitro models we employed in correlation with findings from the literature already mentioned. The metabolizations were conducted with suspension cells as well as microsomes without the addition of serum proteins, so that stabilizing, proteinogenic effects described in the literature, such as the binding of TAM to proteins and/or the non-specific protein binding on surfaces (e.g., due to the Vroman effect), are only partially effective in reducing adsorption in our models. We consider decisive effects on the quantification of TAM due to solubility effects in the in-vitro systems to be unlikely, as precipitation of TAM was not observed in preliminary studies with KHB buffer, cell culture medium (ProCHO5), and in KPO4, as well as during the metabolization experiments. Furthermore, sampling for HPLC-quantification was always conducted from homogeneous solutions, which were mixed 1:1 in acetonitrile prior to further processing, ensuring that potentially precipitated TAM at the microscopic level was re-dissolved.

The aforementioned insights from the literature partially explain our observation  …”

Reviewer’s comment #2-4:

  1. Pre-treating CHO cells with DMSO: Isn’t 2% DMSO too toxic to the cells?

Our response to comment reviewer #2-4:

We thank the reviewer for his question and would like to comment on it.

The pre-treatment of CHO cells with 2% DMSO is carried out for a maximum of 24 h. After this period, a slight cytotoxic effect on the cells can be observed; however, cell viability decreases only marginally. In contrast, CYP activity in the cells or in isolated microsomes from these cells is significantly increased. A pre-treatment of the cells for 48 h results in further reduced viability, but does not lead to any additional increase in CYP activity. Therefore, the pre-treatment is a trade-off between reducing vital cell number and achieving maximum CYP activity. We also found in preliminary experiments that adherent cells exhibit significantly higher tolerance to DMSO compared to suspension cells. With adherent cells, a 24-hour pre-treatment with 3% DMSO was uncritical. The mechanism underlying this observation is difficult to ascertain due to the limited literature available. Our research indicated that DMSO is supposed to enhance DNA accessibility, thereby increasing gene expression in our recombinant system. The strong CMV promoter we used is likely very beneficial in this context. We hope that we were able to answer the reviewer's question satisfactorily.

Round 2

Reviewer 2 Report

Comments and Suggestions for Authors

The authors have adequately addressed my comments.